

# The fence experiment – full-scale lidar-based shelter observations

Alfredo Peña[1], Andreas Bechmann[1], Davide Conti[1], and Nikolas Angelou[1]

[1]DTU Wind Energy, Technical University of Denmark, Roskilde, DK

*Correspondence to:* Alfredo Peña (aldi@dtu.dk)

**Abstract.** We present shelter measurements of a fence from a field experiment in Denmark. The measurements were performed with three lidars scanning on a vertical plane downwind of the fence. Inflow conditions are based on sonic observations of a nearby mast. For fence-undisturbed conditions, the lidars' measurements agree well with those from the sonics and, at the mast position, the average inflow conditions are well described by the logarithmic profile. Seven cases are defined based on the relative wind direction to the fence, the fence porosity, and the inflow conditions. The larger the relative direction, the lower is the shelter. For the case with the largest relative directions, no shelter is observed in the far wake (distances $\gtrsim 6$ fence heights downwind of the fence). When comparing a near-neutral to a stable case, a stronger shelter effect is noticed. The shelter is highest below $\approx 1.46$ fence heights and can sometimes be observed at all downwind positions (up to 11 fence heights). Below the fence height, the porous fence has a lower impact on the flow close to the fence compared to the solid fence. Velocity profiles in the far wake converge onto each other using the self-preserving forms from two-dimensional wake analysis.

## 1 Introduction

The flow around obstacles is difficult to observe and model because of the turbulence characteristics and velocity shears. Such flow has not received much attention in wind energy partly due to the urge to decrease the cost of energy, narrowing the research on flow characteristics to large-turbine operating conditions. These turbines generally operate in areas and at heights, where the obstacles' effects can be neglected. However, due to the decrease of available 'high wind' sites on land, turbines are being deployed in environments, where obstacles cannot be ignored. Also, the 'small' turbine industry has steadily grown (Gsänger and Pitteloud, 2014) and small machines are commonly installed close to obstacles. Due to shelter, such installations often result in lower-than-expected yields and turbine breakdown.

Computational fluid dynamics (CFD) methods, e.g. those solving the Reynolds-averaged Navier-Stokes (RANS) equations, can accurately describe the flow around obstacles and are used to study specific flow conditions (Iaccarino et al., 2003). However, they are often too expensive to be implemented in wind-resource assessment tools. Therefore, the obstacles' effect is normally estimated using 'engineering'-like models. Some, e.g. WEMOD (Taylor and Salmon, 1993) and WAsP-shelter (Mortensen et al., 2007), are based on the analytical theory by Counihan et al. (1974), which describes the wake behind two-dimensional (2D) obstacles.

Analytical theories and CFD simulations, have mainly been evaluated with wind-tunnel data (Castro and Robins, 1977) and few full-scale three-dimensional (3D) shelter experiments have been performed. Nägeli (1953) is perhaps the first to investigate



the mean velocity profiles downwind porous windbreaks, although his data are not of the highest quality (Seginer, 1972). Most shelter experiments are associated with agro-engineering studies, where the purpose is windbreak optimization for stock and crop protection, and are focused on porous obstacles (Nord, 1991).

Here, we present a comprehensive dataset of full-scale measurements of a fence shelter. The measurements were conducted
at Risø's test site in Denmark and the WindScanner (WS) lidar-based system was used to measure the 3D wind vector on a vertical plane. The experiment's objective is to serve as benchmark for shelter models. Section 2 introduces the definitions and theory used to analyze the measurements. Section 3 provides details of the site and the measurements, Sect. 4 describes the way data are analyzed, and Sect. 5 presents the shelter results for a number of inflow conditions/cases. Finally, Sect. 6 provides some discussion and conclusions about the campaign and future model evaluation.

## 2  Definitions

### 2.1  Problem

We want to describe the turbulent flow behind a 2D fence (Fig. 1) and compare it to the undisturbed inflow (subscript $_o$). We use a right-handed Cartesian coordinate system with the three velocity components, $u$, $v$, and $w$, aligned with the $x$, $y$, and $z$ (the vertical) axes, respectively. The horizontal wind-speed magnitude is thus $U = \left(u^2 + v^2\right)^{1/2}$. The coordinate center is
placed on the ground at the fence. The flow is described by the roughness length $z_o$ and the fence height $h$.

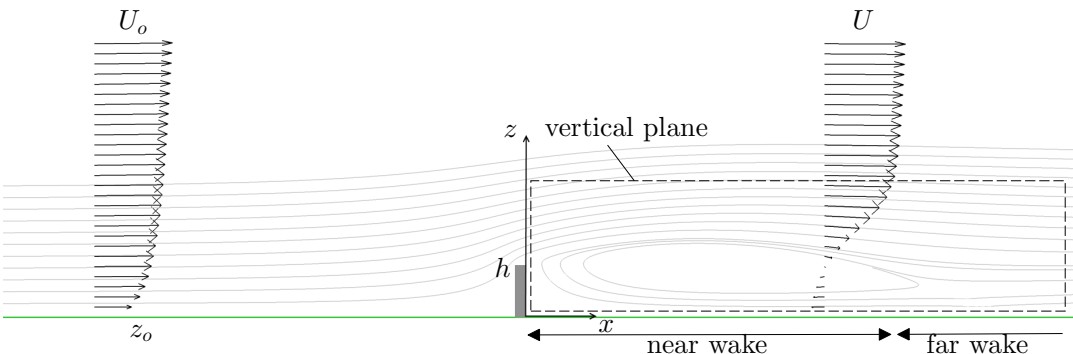

**Figure 1.** Turbulent flow around a 2D fence and the vertical plane of interest

We investigate the flow on a 2D vertical plane extending $2.5\,h$ vertically and $\approx 11\,h$ horizontally downstream of the fence. For simplicity, two main regions are defined in this plane: the 'near-wake' ($x < 6\,h$) and the 'far-wake' ($x > 6\,h$) regions. We will describe the flow for different inflow directions and that perpendicular to the fence (along the $x$-axis).



## 2.2 Inflow

We assume that the inflow can be described by the diabatic wind profile (Stull, 1988),

$$U_o(z) = \frac{u_*}{\kappa} \left[ \ln \left( \frac{z}{z_o} \right) - \psi_m(z/L) \right], \tag{1}$$

where $u_*$ is the friction velocity, $\kappa$ the von Kármán constant ($\approx$0.4), and $\psi_m$ a function of the dimensionless stability parameter $z/L$, being $L$ the Obukhov length. $u_*$ and $L$ can be computed as

$$u_* = \left( \overline{u'w'}^2 + \overline{v'w'}^2 \right)^{1/2}, \tag{2}$$

$$L = -\frac{u_*^3}{\kappa (g/\overline{T}) \overline{w'\Theta_v'}}, \tag{3}$$

where $g$ is the gravitational acceleration, $T$ a reference temperature, $\Theta_v$ the virtual potential temperature, the primes denote fluctuations around the time average, and the overbar a time average.

## 2.3 Two-dimensional wake theory

The speed-up $U/U_o$ at a specific height $z$ is used to quantify the shelter and can be written as

$$\frac{U(z)}{U_o(z)} = 1 - \frac{\Delta U(z)}{U_o(h)} \frac{U_o(h)}{U_o(z)}, \tag{4}$$

where $\Delta U(z) = U_o(z) - U(z)$. The term $\Delta U(z)/U_o(h)$ is predicted by Counihans et al.'s analytical theory, in which a 2D obstacle wake is divided into three regions. Within the mixing region, spreading from the obstacle's top, the velocity is self-preserving with the form,

$$\frac{\Delta U(z)}{U_o(h)} = \frac{C/I(n)}{K\, h^2\, U_o(h)^2} \left( \frac{x}{h} \right)^{-1} \frac{d}{d\eta} \left[ \eta^2 \,_1F_1 \left( \frac{2-n}{2+n}, \frac{n+4}{2+n}, \frac{-\eta^{n+2}}{(n+2)^2} \right) \right], \tag{5}$$

where $C$ is related to the wake strength (see below), $K = 2\kappa^2/\ln(h/z_o)$, $n$ the inflow's shear exponent, $_1F_1$ the confluent hypergeometric function, $\eta$ a dimensionless length scale related to the mixing-region depth, and $I$ an integral constant for the wake's self-preserving solution in the mixing region. The latter two are expressed as

$$I(n) = \frac{(1+n)(2+n)^{(4+n)/(2+n)}}{1+2n} \frac{\Gamma\left(\frac{4+n}{2+n}\right)\Gamma\left(\frac{1-n}{2+n}\right)}{\Gamma\left(\frac{2-n}{2+n}\right)}, \quad \text{and} \tag{6}$$

$$\eta = \left( \frac{z}{h} \right) \left[ \frac{K\, x}{h} \right]^{-1/(n+2)}. \tag{7}$$

Counihan et al. showed that profiles of $\frac{\Delta U(z)}{U_o(h)}\left(\frac{x}{h}\right)$ as function of $\eta$ converge onto each other within the far-wake region $6 \leq x/h \leq 30$ from full-scale measurements of the wind behind porous windbreaks and within the range $7.5 \leq x/h \leq 72$ from





wind-tunnel measurements. Based on Counihan et al.'s theory and using wind-tunnel measurements behind 2D fences, Perera

(1981) proposed an expression that became the basis of engineering obstacle models,

$$\frac{\Delta U(z)}{U_o(h)} = A (1 - \varphi) \left(\frac{x}{h}\right)^{-1} \eta \exp\left(-0.67\eta^{1.5}\right),$$ (8)

where $\varphi$ is the fence porosity and $A$ a constant ($= 9.75$).

The solution to the term $\frac{d}{d\eta}[...]$ in Eq. (5) is unattractive but for the special case $n = 0$, it is simple ($= 2\,\eta \exp\left(-0.25\eta^2\right)$). The self-similar profile $\frac{\Delta U(z)}{U_o(h)}\left(\frac{x}{h}\right)$ shows a maximum at $\eta(z/h \approx 1)$ and approaches zero with increasing $\eta$. For decreasing $n$

values, the zero approach occurs at smaller $\eta$ values and the profile's maximum slightly decreases (only 7% between $n = 0.14$ and 0). Also, $I$ is not that sensitive to $n$ ($= 7.64$ and $7.08$ for $n = 0.14$ and 0, respectively), and $C = C_h\,h^2 U_o(h)^2$ from pressure measurements on blocks in shear flows. Therefore, Eq. (5) can be simplified to

$$\frac{\Delta U(z)}{U_o(h)} = \frac{C_h}{K\,I(n = 0.14)} \left(\frac{x}{h}\right)^{-1} 2\,\eta \exp\left(-0.25\eta^2\right).$$ (9)

Counihan et al. chose $C_h = 0.8$ for measurements behind 2D blocks. Following the analysis by Taylor and Salmon (1993),

$C_h$ corresponds to the wake-moment coefficient. They suggest $C_h = B(1 - \varphi)$ with $0.2 \leq B \leq 0.8$ depending on the obstacle type.

## 3 Site and measurements

We aim at describing the effect of a full-scale obstacle on the atmosphere by measuring on a vertical plane downwind of a fence. Here, we first describe the site, the inflow conditions from mast measurements, and the shelter measurements performed

by the WS.

### 3.1 Site

The 'fence experiment' took place at Risø's test site, which is $\approx 7$ km north from Roskilde and $\approx 35$ km west from Copenhagen, Denmark (Fig. 2). It was conducted during two periods: from March 10 to April 1 the fence was solid and from September 29 to October 2 2015 the fence was made porous. The terrain at the site is slightly hilly and the surface is characterized as a mix

between cropland, grassland, artificial land, and coast.

The fence was made of horizontal wooden panels with wooden beams on each side supporting the structure (see Fig. 3-bottom frames). For the second period of the experiment, the fence porosity (ratio of the 'pores' to the total area) is 0.375. The fence is 3-m high, 30-m wide, and 0.04-m thick (the wooden vertical poles are 0.1-m thick). The center point of the fence has coordinates 694477.5E, 6175332N (UTM32 WGS32) and is $\approx 78$ m southeast of the Roskilde Fjord coastline. Due to land

restrictions and the orientation of the coastline, the fence is oriented $\approx 42°$ from the true north (winds from the direction $\approx 312°$ are normal to the fence).

The terrain's slope behind the fence was measured with a Trimble global positioning system (GPS), along two lines from its corners. Figure 4-top illustrates the fence experiment and the instrumentation. Figure 4-bottom illustrates the positions where





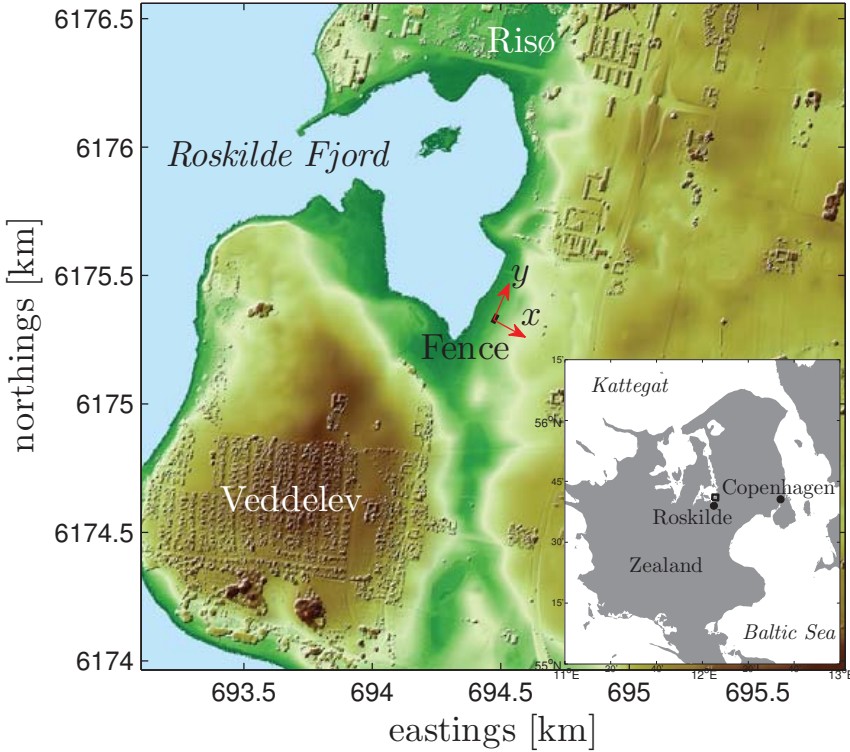

**Figure 2.** The fence experiment on a digital surface model (UTM32 WGS84) of the area surrounding Risø's test station. Cropland and grassland are shown in green, cropland and artificial land in light brown, rural areas and buildings in brown, and the waters from the fjord in light blue. The reference coordinate system is shown in red. In the bottom-right part, the test site location (black rectangle) on the island of Zealand, Denmark, is illustrated

we want to measure on the vertical plane (described in Sect. 3.3) and the terrain elevation. Note that the reference system is not at the fence center but 1.53 m southwest and so fence corners are not at the same distance from the reference system (see Table 1). The terrain height above the fence base for the positions at which we want to measure the shelter is provided at http://www.fence.vindenergi.dtu.dk. The relative direction to the fence, $\theta$, is defined positively increasing clockwise.

### 3.2 Meteorological mast

A mast is deployed northeast of the fence and two Metek USA-1 sonic anemometers are placed on booms oriented towards the fence at 6 and 12 m above the ground and record time series of the three wind speed components and temperature at 20 Hz. Mean and turbulence statistics are estimated over 10-min periods from the sonic measurements (we also analyze the sonics' time series in shorter time periods as described in Sect. 4). The sonics' times series are linearly detrended over the 10-min period, and mean and turbulence quantities like $u_*$ and $L$ are estimated from the 10-min statistics. The terms $T$ and $\overline{w'T_v'}$ in





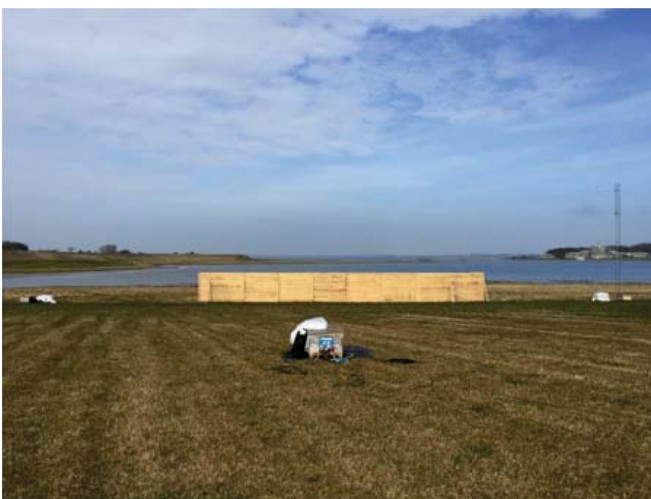

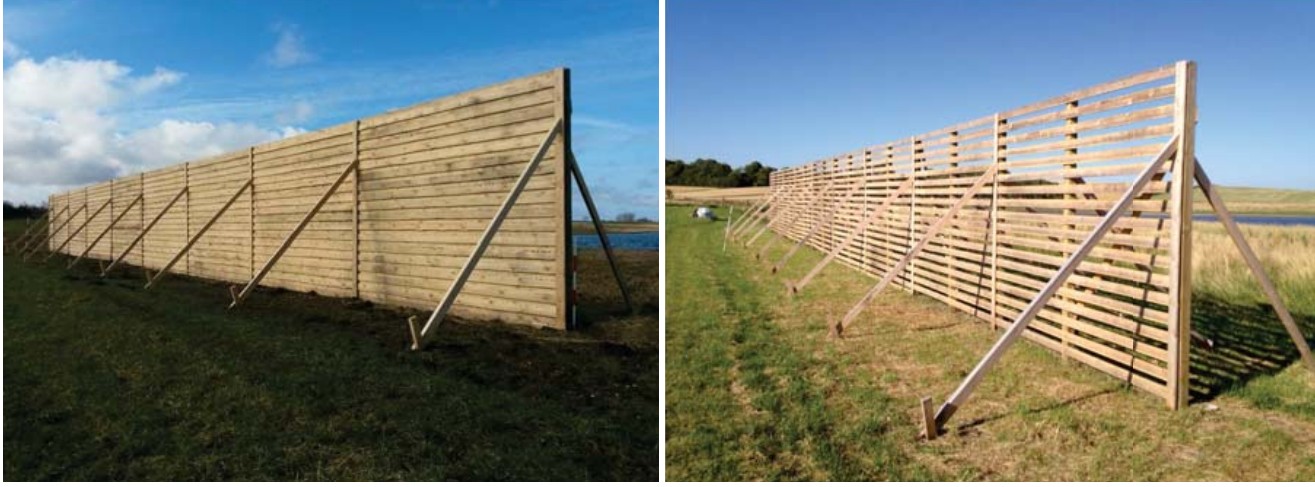

**Figure 3.** Photographs of the fence experiment. (Top) overview of the fence and the instrumentation including the lidars and the mast. (Left) solid and (right) porous fence setups

Eq. (3) are estimated from the sonics' temperature and kinematic heat flux, respectively. For the latter, we use the crosswind corrections of Liu et al. (2001).





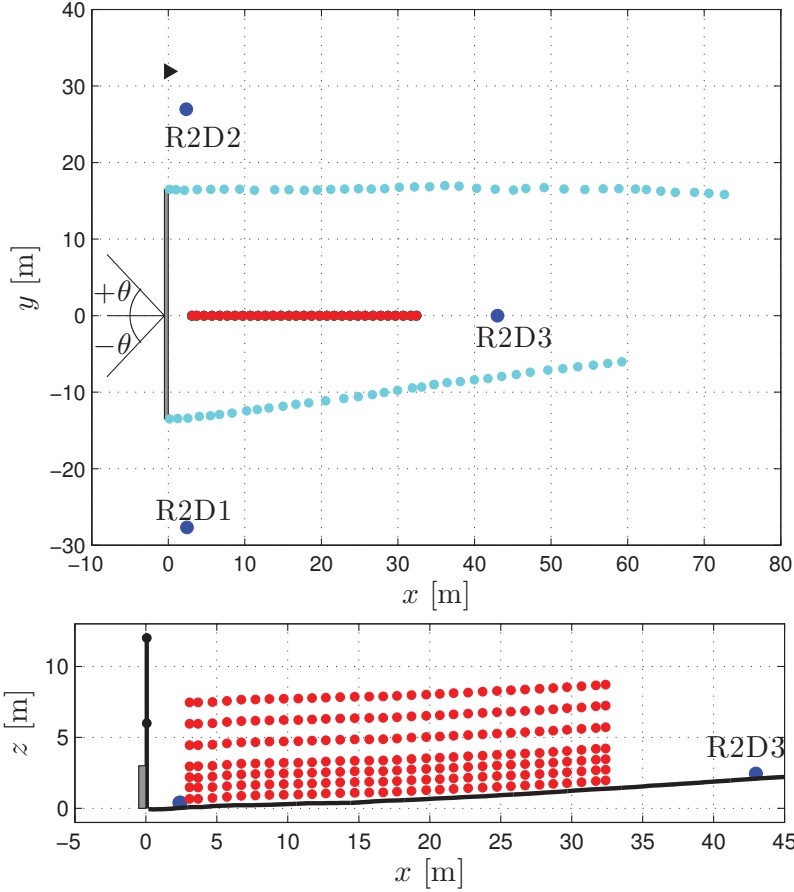

**Figure 4.** The fence experiment in the reference coordinate system. The positions of the fence (gray rectangle), the lidars (blue circles), the mast (black triangle and black thick line), scanning grid (red circles), and GPS measurements (cyan circles) are also illustrated both at the top (top) and side (bottom) views. The terrain elevation is also shown in the side view

## 3.3 Lidar measurements

### 3.3.1 Basics

5    The three velocity components on the vertical plane are measured using three short-range lidars that are synchronized both in time and space. These three devices conform the WS. The instruments are based on a continuous-wave coherent lidar (Karlsson et al., 2000), which is capable of measuring the radial (or line-of-sight) speed and its direction (Sjöholm et al., 2014).

     The lidars do not perform point-like but volume measurements. The volume depends on the probe length of each lidar, which is considered to be twice the Rayleigh length $z_R$. At focused distances of 28 and 42 m, the lidars operate with $z_R = 0.67$ and

5    1.52 m, respectively (Sonnenschein and Horrigan, 1971).





### 3.3.2 WindScanner simulation

An optimized positioning of the lidars is a compromise between the size of the scanned area, the error in wind speed (which increases with the size of the scanned area), and the wind speed components (which we are most interested in accurately measuring). As one lidar measures the line-of-sight velocity only, we need to deploy at least one as far downwind as possible, so that under 'experimentally-ideal' inflow conditions ($\theta \approx 0°$) this unit measures most of the $u$-component, and as close to the fence to avoid interference of the probe volume with the fence itself.

A CFD solver of the RANS equations (EllipSys) (Sørensen, 2003) with a standard $k$-$\varepsilon$ model was used to simulate the flow behind the fence (the solid setup only) and the CFD results were used to 'simulate' the flow field observed by the WS including the effect of the probe volume. The CFD simulation was performed using flat terrain with $h/z_o = 300$. A logarithmic profile in balance with the ground roughness was used as inlet condition for $\theta = 0°$. To correctly model the high near-fence velocity gradients, the CFD grid had a 0.03-m wall resolution, which was coarsened with distance to the wall. CFD results

were extracted from the same vertical plane as scanned by the WS.

    Figure 5 shows both the CFD and the WS's simulated flow assuming that the CFD results 'follow' the terrain elevation. The largest differences for the $u$-component occur close to the fence and at $z/h = 1.50$ but the relative error is highest for the vertical levels close to the ground. Similarly for $w$, the difference generally increases the closer to the fence and is highest at the two first vertical levels. These are the areas where the CFD simulation results show the highest $w$-gradients and so we

expect to have large uncertainty in the $w$-measurements by the WS. A number of positions were tested and the one shown in Fig. 4 and Table 1 was selected because it gave the lowest error for both the $u$- and $w$-components 'simulated' by the WS when compared to the CFD results.

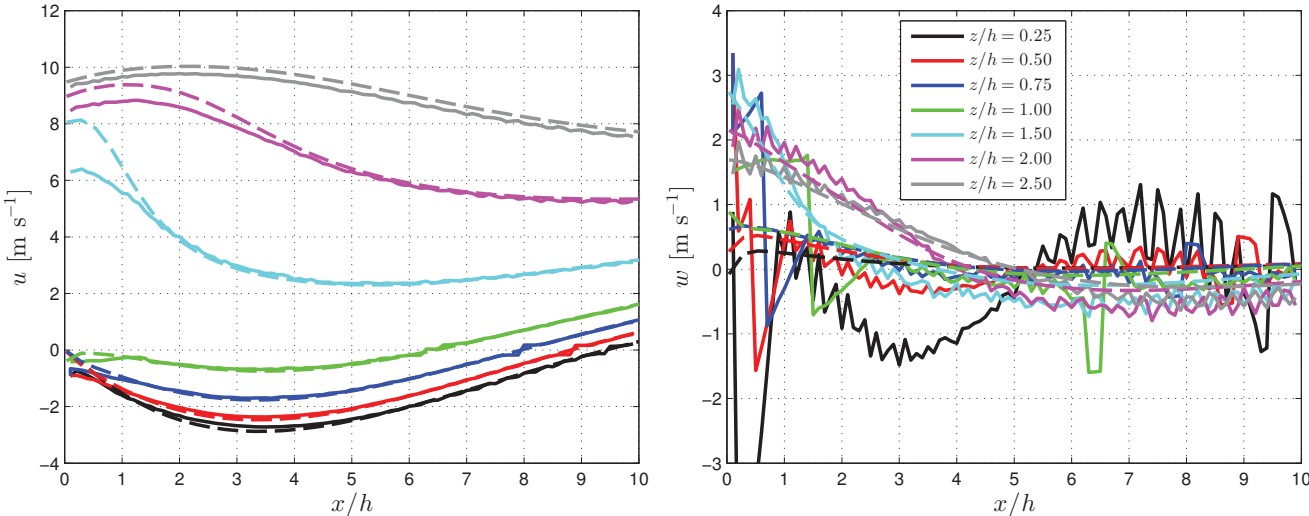

**Figure 5.** Simulation of the WS measurements (solid lines) and the CFD simulation results (dashed lines) for $u$ (left frame) and $w$ (right frame) and several vertical levels



**Table 1.** Instrumentation coordinates for the fence experiment

| Instrument | $x$ [m] | $y$ [m] | $z$ [m] |
|---|---|---|---|
| R2D1 | 2.43 | -27.67 | 0.40 |
| R2D2 | 2.36 | 26.96 | 0.40 |
| R2D3 | 43.00 | 0 | 2.43 |
| Sonics | 0.06 | 31.91 | 6, 12 |
| Fence (southern corner) | 0 | -13.51 | 0, 3 |
| Fence (northern corner) | 0 | 16.53 | 0, 3 |

### 3.3.3 Experimental details

The scanning pattern on the vertical plane was decided based on the CFD simulation results and the regions where we are interested in measuring the shelter; we wanted to measure in both the near- and far-wake regions, and below and above $h$ (up to $z/h \approx 2.5$). The WS's lidars were therefore set to synchronously scan from a position 1 m downwind the fence up to a distance of 10 $h$ and at 7 different levels following the terrain elevation. The lidars were continuously acquiring line-of-sight velocity spectra at $\approx$49 Hz. The spectra were gridded in 1-m cells and spatially averaged in each cell leading to 31 space- and time-averaged spectra per line. The final scanning grid has thus $31 \times 7$ points in the $x$-$z$ plane. The 7 vertical levels are at the heights $[0.21, 0.46, 0.71, 0.96, 1.46, 1.96, 2.46] \, h$. The 31 positions along the $x$-axis are given at http://www.fence.vindenergi.dtu.dk.

A 'full-scan' (a complete measurement of all 217 grid positions), took for most days of the campaign $\approx$21 s. During the second period of the campaign, one lidar had problems with the focus mechanism and to increase the amount of full-scans, we redefine the full-scan on a smaller grid of $29 \times 7$ points, i.e. excluding the grid points furthest and closest to the fence.

After the line-of-sight spectra are averaged in each cell, a series of post-processing steps were performed to first remove noise signals and, subsequently, a median frequency estimator was applied to derive the line-of-sight velocity in each spectrum as in Angelou et al. (2012). The minimum detectable speed of the WS is $\approx$0.15 m s$^{-1}$; thus, when the line-of-sight velocity is lower than this value, the WS reports a zero line-of-sight velocity. We filter out full-scans where line-of-sight velocities are zero or appear as peaks in the time series (for the latter using the method by Goring and Nikora, 2002) for each lidar and grid position.

The $u$-, $v$-, and $w$-components are estimated at each grid position from the scan geometry combined with the line-of-sight velocities. A preliminary analysis of the estimations of $w$ at the first two vertical levels showed unrealistic values because the line-of-sight of the lidars is almost perpendicular to $w$. Therefore, for all the positions in these two levels, we use the line-of-sight velocities of R2D1 and R2D3 only, so at these two levels we can only estimate $u$ and $v$.

The WS was mostly operated when the sonics indicated westerlies and during periods without rain. The WS measurements are thus concentrated on few days as indicated in Table 2, which shows the amount of full-scans per day.



**Table 2.** Number of full-scans per day by the WS and the fence porosity

| Date | No. of full-scans | porosity |
|---|---|---|
| March 10 | 637 | solid |
| March 11 | 712 | solid |
| March 20 | 11 | solid |
| March 26 | 84 | solid |
| March 27 | 81 | solid |
| April 01 | 27 | solid |
| September 30 | 11 | porous |
| October 01 | 107 | porous |
| October 02 | 125 | porous |

## 4   Data analysis

### 4.1   Sonic-lidar intercomparison

Besides the 10-min mean and turbulence sonic statistics, we derive another set of sonic statistics based on the time period that the WS takes to complete each full-scan (denoted by a $^\sim$ symbol). Thus, we also know both the mean wind speed and direction, and their variability, within this shorter period.

The grid point closest to the sonic at 6 m is at a height of $\approx$6 m. We compare the WS measurements at this grid position with those from the 6-m sonic. This is not a fair comparison because the measurements from the WS at each grid position are nearly 'instantaneous', i.e. it takes less than 0.1 s to scan each grid point (we will use a $^\wedge$ symbol to refer to them), whereas we use the full-scan period for the sonic measurements. However, the comparison will show us the conditions in which the flow at both positions is similar. Fig. 6 shows a scatter plot of such measurements for both periods of the campaign.

Figure 6-left illustrates the good agreement between the 6-m sonic and the WS for the horizontal wind speed magnitude; the scatter is low and high for low and high wind speeds, respectively. Figure 6-right shows that the degree of scatter is a function of the relative wind direction; when the WS measures downwind the fence ($|\widetilde{\theta}_{\text{sonic}}| \leq 90°$), the scatter is much higher than for upwind conditions. Although the grid point used is $\approx$1 $h$ above and $\approx$1 $h$ downwind the fence, there seems to be a strong effect of the fence on the flow at this position, whereas the effect is nearly negligible for $|\widetilde{\theta}_{\text{sonic}}| \geq 90°$. A similar analysis for downwind conditions using the grid point furthest away from the fence ($\approx$32 m) and at the same vertical level shows a reduction of the scatter (not shown) as the shelter is low there (see Sect. 5). Figure 6-right also shows that most of the measurements are concentrated at $\widetilde{\theta}_{\text{sonic}} \approx -50°$ and that few winds are normal to the fence.





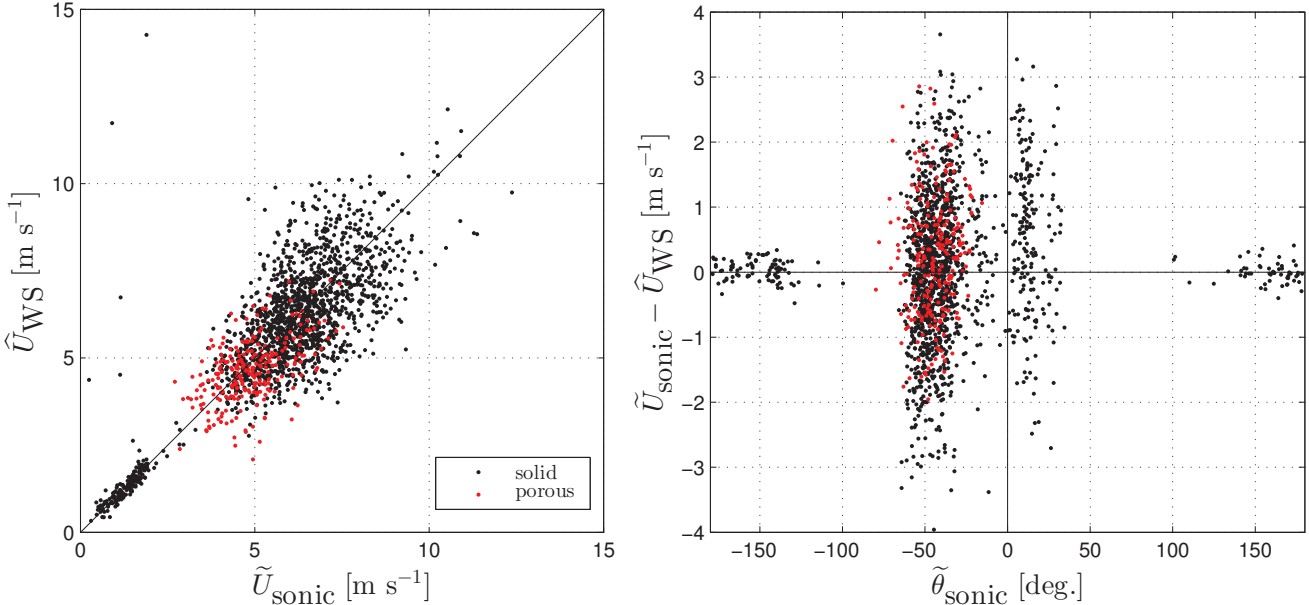

**Figure 6.** (Left) scatter plot of wind speed measurements from the 6-m sonic and the WS for the grid point closest to the fence and at height of $\approx 6$ m. (Right) the difference between these two measurements as function of the relative wind direction observed by the 6-m sonic

### 4.2 Inflow conditions

The flow at the mast position (assumed to be undisturbed by the fence for $-75 \leq \theta \leq 75°$) determines the inflow conditions required to estimate the speed-up due to shelter. Therefore, we need to estimate the surface conditions as a function of relative wind directions at the site.

     Assuming homogenous inflow over flat terrain, we estimate $z_o$ from Eq. (1) using 10-min mean and turbulence sonic statistics. Thus, we have two $z_o$-values derived using either sonic for each 10-min period. To compute $\psi_m$, we use the forms in

Peña (2009). Table 3 shows the median of such $z_o$-estimations based on the 6- and 12-m sonics for March and September 2015 and for $10°$ $\theta$-intervals (we use the 10-min mean sonic wind direction $\overline{\theta}_{\rm sonic}$ to classify the 10-min sonic statistics into relative direction intervals). As shown, for both periods, $z_o$ increases with increasing $|\theta|$, as expected, due to the topography upstream the fence (see Fig. 2). Further, the difference in $z_o$-values between both periods is relatively small indicating that, particularly at $\theta \approx 0°$, $z_o$ is greatly influenced by the fjord's surface conditions.

We need to find out if it is sufficient to describe the inflow using Eq. (1) with the $z_o$-values in Table 3 given that the terrain is not flat, the upstream conditions not homogeneous, and the atmospheric conditions generally not neutral. For this, we define 'case' studies based on $\theta$-intervals and we select the data, which are included in each case, using the full-scan mean direction from the 6-m sonic, i.e. $\widetilde{\theta}_{\rm sonic}$ (see Table 4).



**Table 3.** Roughness length $z_o$ as function of the relative wind direction $\theta$ based on either the 6- or the 12-m sonic for both March and September 2015. The amount of 10-min samples is also shown

| | March | | | September | | |
|---|---|---|---|---|---|---|
| $\theta \pm 5°$ | 6-m $z_o$ [m] | 12-m $z_o$ [m] | No. of 10-min samples | 6-m $z_o$ [m] | 12-m $z_o$ [m] | No. of 10-min samples |
| -90 | 0.0673 | 0.0785 | 176 | 0.0549 | 0.1204 | 184 |
| -80 | 0.0435 | 0.0542 | 174 | 0.0280 | 0.0574 | 193 |
| -70 | 0.0231 | 0.0173 | 174 | 0.0095 | 0.0143 | 155 |
| -60 | 0.0095 | 0.0070 | 116 | 0.0072 | 0.0089 | 95 |
| -50 | 0.0069 | 0.0095 | 114 | 0.0052 | 0.0068 | 79 |
| -40 | 0.0049 | 0.0075 | 148 | 0.0028 | 0.0048 | 120 |
| -30 | 0.0031 | 0.0051 | 183 | 0.0012 | 0.0024 | 108 |
| -20 | 0.0031 | 0.0036 | 61 | 0.0009 | 0.0014 | 70 |
| -10 | 0.0021 | 0.0033 | 40 | 0.0004 | 0.0005 | 74 |
| 0 | 0.0014 | 0.0010 | 33 | 0.0009 | 0.0018 | 118 |
| 10 | 0.0013 | 0.0026 | 61 | 0.0018 | 0.0064 | 140 |
| 20 | 0.0014 | 0.0051 | 15 | 0.0030 | 0.0046 | 106 |
| 30 | 0.0015 | 0.0043 | 43 | 0.0060 | 0.0084 | 93 |
| 40 | 0.0020 | 0.0038 | 16 | 0.0121 | 0.0077 | 47 |
| 50 | 0.0113 | 0.0447 | 10 | 0.0407 | 0.0536 | 15 |
| 60 | 0.0280 | 0.0859 | 12 | 0.0975 | 0.2840 | 11 |
| 70 | 0.0204 | 0.0778 | 34 | 0.0172 | 0.0641 | 1 |
| 80 | 0.0149 | 0.0970 | 55 | 0.0151 | 0.1155 | 10 |
| 90 | 0.0330 | 0.2586 | 73 | 0.0289 | 0.3818 | 7 |

**Table 4.** Case studies for a number of $\theta$-intervals. Refer to the text for details

| Case | Porosity | $\theta$ [deg.] | $\langle z_o \rangle$ [m] | $u_{*est}$ [m s$^{-1}$] | $\langle z/L \rangle$ | No. of full-scans |
|---|---|---|---|---|---|---|
| I | solid | $0 \pm 15$ | 0.0016 | 0.36 | 0.021 | 159 |
| II | solid | $0 \pm 30$ | 0.0019 | 0.36 | 0.015 | 304 |
| III | solid | $-30 \pm 15$ | 0.0037 | 0.34 | 0.023 | 604 |
| IV | solid | $-60 \pm 15$ | 0.0131 | 0.39 | 0.045 | 583 |
| V | solid | $30 \pm 15$ | 0.0016 | 0.35 | 0.007 | 62 |
| VI | solid | $0 \pm 30$ | 0.0019 | 0.28, 0.27 | 0.044 | 92 |
| VII | porous | $-30 \pm 15$ | 0.0016 | 0.25 | -0.068 | 128 |



The case studies are selected so that each has a significant number of full-scans and that we can study the influence on the shelter of a wider $\theta$-interval (cases I and II), $\theta$ itself (cases I and III–V), atmospheric stability (cases II and VI), and porosity (case VII). Table 4 provides an estimation of different parameters that are used to reproduce the inflow conditions for each case, which are also illustrated in Fig. 7-left. For each case, we:

1. ensemble-average the $z_o$-values in Table 3 within the $\theta$-interval in Table 4 (we denote ensemble averages with the $\langle \rangle$ symbol),

2. estimate a 'new' friction velocity $u_{*\mathrm{est}}$ with Eq. (1) assuming $\psi_m(z/L) = 0$ and using the sonic wind speed measurement at 6 m, ensemble-averaged from the sonic mean wind speeds within the full-scan period,

$$u_{*\mathrm{est}} = \frac{\kappa \, \langle \widetilde{U} \rangle_{\mathrm{sonic}}}{\ln\left(6 \text{ m}/\langle z_o \rangle\right)},\tag{10}$$

3. estimate the 'mean' inflow wind profile $\langle U_o(z) \rangle$[1] using Eq. (1) assuming $\psi_m(z/L) = 0$ (solid color lines in Fig. 7-left) as

$$\langle U_o(z) \rangle = \frac{u_{*\mathrm{est}}}{\kappa} \ln\left(\frac{z}{\langle z_o \rangle}\right).\tag{11}$$

As shown, the estimations of the inflow profiles are in good agreement with the sonic measurements (an absolute error of 0.18 m s$^{-1}$ is computed at 12 m for case V as the largest of all cases). We therefore assume that, although present, the topographic effects at the mast position within the heights 6–12 m can be neglected for these $\theta$-ranges. The inflow is thus well described by the logarithmic profile.

In addition, Fig. 7-left shows three more profiles for case I. The black dashed line shows the mean inflow conditions but using the ensemble-average $u_*$ of $u_*$-values estimated from the 6-m sonic with Eq. (2) within the full-scan period, i.e. $\langle \widetilde{u_*} \rangle$. In this case, there is a systematic underestimation of the inflow wind speed because $\langle \widetilde{u_*} \rangle$ is about $13\%$ lower than $u_{*\mathrm{est}}$ (the latter is given in Table 4). The results in the black dash-dotted line are obtained similarly to those in the solid lines but using the 12-m sonic and the $\langle z_o \rangle$ derived from the observations at that height. Therefore, the estimated inflow wind speed at 12 m is equal to the ensemble-average sonic wind speed at the same height. The results in the black dotted line are found with the same methodology as that used for the results in the dashed line but with the 12-m sonic. From these three results, we confirm: first, that turbulent fluxes estimated in the short period of the full-scan are not adequate for deriving the inflow conditions (see the work of Lenschow et al., 1994) and, second, that similar results are obtained when using $z_o$-estimations based on either the 6- or 12-m sonic. This also gives us an idea of the small effect that the internal boundary layer (developed at the coastline) has on the inflow profile at the mast position and within the heights between the sonics.

For case VI, a second mean inflow profile (magenta dashed line) is shown in Fig. 7-left. Case VI is similar to case II but we narrow the analysis to stable conditions $z/L \geq 0.01$ from the 'concurrent'[2] 10-min derived turbulence sonic estimates at 6 m. $u_{*\mathrm{est}}$ can be computed as in Eq. (10) and, in addition, the correction due to atmospheric stability can be included (the result is

---

[1]Although this is not an ensemble average per definition, we use the $\langle \rangle$ symbol because it results from the ensemble-averaged roughness length $\langle z_o \rangle$

[2]Quotation marks because a full-scan take less than 10 min



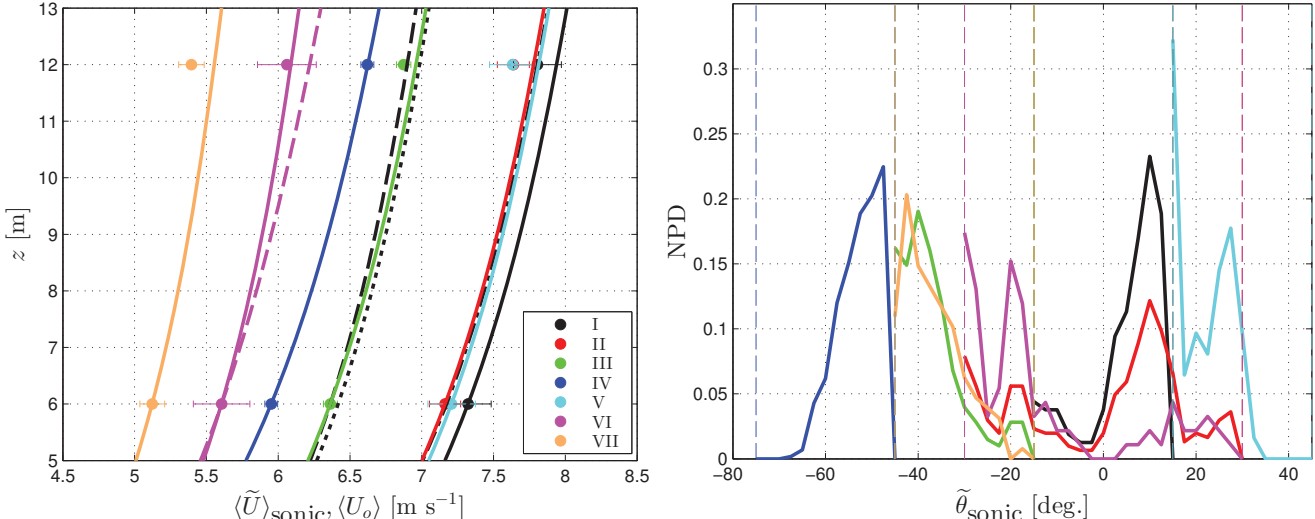

**Figure 7.** (Left) Inflow conditions for the case studies. The circle markers indicate the ensemble-averaged sonic measurements $\langle \widetilde{U} \rangle_{\text{sonic}}$ ($\pm$ the standard error in the error bar) and the lines the estimations of the mean inflow conditions $\langle U_o(z) \rangle$ (see text for details). (Right) normalized distribution (NPD) of the relative wind direction from the 6-m sonic $\widetilde{\theta}_{\text{sonic}}$ for the case studies

the second value for the $u_{*\text{est}}$ column in Table 4). Thus, the magenta dashed line shows the mean inflow profile using Eq. (11)

5 with this new $u_{*\text{est}}$ value, which overestimates the mean wind speed at 12 m by 0.16 m s$^{-1}$ only.

For each case in Table 4, we include the average dimensionless stability $\langle z/L \rangle$-value, which is found by ensemble-averaging the 10-min turbulence fluxes from the 6-m sonic that are 'concurrent' with the time of the full-scans. As shown, the atmosphere for the 'solid fence' cases is in average stable, except for case V, which corresponds to the most northern winds, and for the 'porous fence' case the atmosphere is unstable. Interestingly, although we do not narrow the filtering criteria to stable conditions

10 for case IV, $\langle z/L \rangle$ is higher for this case than for case VI.

Figure 7-right shows that the distribution of $\widetilde{\theta}_{\text{sonic}}$-values for each case is not uniform and that the center of the interval, in most cases, differs from the mean of the relative directions within the interval; thus these distributions should be taken into account when evaluating models. We provide the values of such distributions at http://www.fence.vindenergi.dtu.dk.

## 5 Results

### 5.1 Speed-up

We classify the data from the WS's full-scans into the cases in Table 4 using the $\widetilde{\theta}_{\text{sonic}}$-values. The horizontal velocities from the WS are then ensemble-averaged within each case, $\langle \widetilde{U}(z)_{\text{WS}} \rangle$, and the speed-up is estimated by normalizing these averages by the case-correspondent 'mean' inflow profile (described in Sect. 4.2).





The plots in Fig. 8 illustrate the speed-up for each case. Although $\widetilde{\theta}_{\mathrm{sonic}}$ does not uniformly distribute within the chosen
relative direction intervals, the effect of the fence on the flow for varying $\theta$-values is well observed, particularly from the
results between cases I, III and IV (three left frames from the top). Case I, as expected, shows the deepest shelter effect of
these three cases, which diminishes when increasing $|\theta|$ and, for case IV, the effect of the fence is only noticed for $x/h \lessapprox 3$.
For case II, which is defined similar to case I but for a broader $\theta$-interval, the speed-up is smoother and slightly deeper than
that for case I but the differences are not large. This is most probably due to the concentration of full-scans at $\widetilde{\theta}_{\mathrm{sonic}} \approx 10°$ in
both cases. Case VI, the 'stable' case II, also shows a similar behavior but with slightly deeper shelter effects than case II. Case
V, similarly defined as case III but with $\theta$ centered at $30°$, shows reductions up to 50% for $x/h \lessapprox 4$ as case III also does. Case
VII, which is comparable to case III but for a different porosity, does not show speed-ups close to zero but the shelter seems to
extend further away from the fence.

For cases I–III and VI we notice a small region where the speed-up is larger than one, located at $x/h \approx 2.5$ and $z/h \approx 2.5$.
High speed-ups within the range $1 \leq x/h \leq 4$ are also observed for some of the other cases in Figs. 9 and 10. These figures
illustrate the behavior of the speed-up (taking into account the sign of $u$) with distances downwind the fence, for the seven
different levels, and for the seven cases. In Fig. 9, the high speed-up is clearly observed in the results for case IV for nearly all
vertical levels, and for $z/h = 2.46$ (Fig. 10), it is visible for all cases where the fence is solid. In Fig. 11, we show this high
speed-up from the CFD simulation (used to estimate the WS's error in Sect. 3.3). The CFD simulation was performed over
flat terrain without roughness changes and so it is the fence itself what causes the increased vertical velocity shear. Further,
the results in Fig. 9 for case IV, in which the fence has the smallest effect on the vertical plane for $z/h \geq 0.71$, show that the
speed-up is $\approx 1$ for $x/h \geq 7$. This shows us that the effect of the topography on the flow is small at all scan positions on the
vertical plane relative to that at the mast position.

In Fig. 8, the direction and magnitude of the ensemble-averaged WS's $u$-component is also illustrated. A region of reverse
flow is visible for all cases when the fence is solid. This region is also shown in Fig. 11 but for the CFD results it extends much
further downwind because the simulation is performed for $\theta = 0°$ only.

The results in Figs. 9 and 10 confirm those in Fig. 8; for the solid setup, case VI generally shows the highest shelter in the far
wake, systematically followed by cases II, I, III, V, and IV, as expected, due to the relative inflow directions. Interestingly, the
shelter's behavior for case VII follows that of cases I, II and VI in the far wake, does not strongly vary below $h$ in the near-wake
region, and is the only case without reverse flow. For all the other cases, reverse flow can be distinguished and vanishes only at
$z/h \geq 0.96$. The speed-up behaviour with distance from the fence is similar for cases I, II and VI with the largest differences
at $z/h = 0.21$ and $x/h \leq 5$, where the reverse flow peaks in magnitude.

Cases III and V have a similar speed-up behavior; case V systematically showing less shelter and so the differences are most
probably due to the different distribution of $\theta$-values. The average speed-up as function of distance from the fence and for each
level and case are presented at http://www.fence.vindenergi.dtu.dk.





**Figure 8.** Averaged speed-up $\langle \widetilde{U}(z)_{\mathrm{WS}}\rangle / \langle U_o(z)\rangle$ behind the fence for a number of cases. Vectors indicate the magnitude and direction of the ensemble-averaged $u$-component

## 5.2 Self-preserving velocity profiles

5 Using observations from three of cases in which the $\theta$-interval is center at $0°$, we compute the self-preserving forms (Sect. 2) and illustrate them (Fig. 12). We:

1. estimate a 'mean' shear exponent $\langle n\rangle$ using the case-concurrent ensemble-average sonic measurements and the power law,

$$\langle n\rangle = \frac{\ln\left[\langle\widetilde{U}\rangle_{\mathrm{sonic}}(z=6\,\mathrm{m})/\langle\widetilde{U}\rangle_{\mathrm{sonic}}(z=12\,\mathrm{m})\right]}{\ln(6\,\mathrm{m}/12\,\mathrm{m})}, \tag{12}$$





2. compute a 'mean' $K$ using the average roughness (Table 4), $\langle K \rangle = 2 \, \kappa^2 / \ln (h / \langle z_o \rangle)$,

3. use the estimations of the mean inflow (Eq. 11) at the vertical levels and at $z = h$ to compute the average self-similar profiles,

$$\frac{\langle \Delta U(z) \rangle}{\langle U_o(h) \rangle} \left( \frac{x}{h} \right) = \frac{\langle U_o(z) \rangle - \langle \widetilde{U}(z) \rangle_{\text{WS}}}{\langle U_o(h) \rangle} \left( \frac{x}{h} \right), \tag{13}$$

4. estimate a 'mean' $\eta$-value, $\langle \eta \rangle$, based on Eq. (7) using $\langle K \rangle$ and $\langle n \rangle$.

Figure 12 shows the self-preserving profiles for a number of downwind distances; near-wake profiles ($x/h < 5.6$) in grey markers and far-wake profiles ($x/h > 6.24$) in non-grey circles. Equation (8) with $A = 9.75$ is also shown. Further, we fit Eq. (9) to the far-wake profiles ($C_h$ is estimated in a least-squares sense).

The profiles in the near wake do not generally converge onto each other, whereas those in the far wake do, particularly for cases II and VI with the broad direction interval. Equation (9) with the adjusted $C_h$ agrees better with the profiles compared with Eq. (8), particularly where the term $[\langle \Delta U(z) \rangle / \langle U_o(h) \rangle] (x/h)$ peaks (vertical levels below $h$), due to the low $C_h$. For these cases, Perera's estimations result in a general overestimation of the speed-up below $h$.

The adjusted $C_h$-value in Eq. (9) changes considerably for these cases. For the narrow direction interval (case I) it is nearly half the value recommended by Taylor and Salmon for 2D fences and increases the broader the interval. The increase of $C_h$ in case II compared to case I can be explained by the $\theta$-distribution in Fig. 7-right; the ensemble-average relative direction in case I is $6.27°$ and in case II is $0.39°$, which partly explains the larger effect of the fence on the flow for case II. However, the effect on the flow is larger in case VI with an average relative direction of $-12.70°$; thus here the stable conditions might be responsible for the increase in $C_h$ and the deeper wake.

## 6  Conclusions and discussion

Full-scale flow measurements on a vertical plane behind a fence are presented. The measurements were conducted by the WS and agree well with sonic measurements from a nearby mast when the wind is not largely disturbed by the fence. Simulation of the WS measurements reveals that the WS tends to underestimate the magnitude of $u$ at $x/h \lesssim 4$ and $z/h > 1$. This is mostly due to the combination of the high vertical velocity gradient and the large probe volume of the lidar furthest downwind from the fence.

The speed-up depends on the inflow conditions. We assume the topographic effects at each of the positions on the vertical plane to be similar to those at the mast position at the same height, as the speed-up approaches one for the case where the fence effect is lower on the flow (case IV) at $x/h \gtrsim 6$ and for all vertical levels. Between the sonics (6–12 m), the inflow conditions are well described by the logarithmic profile using direction-dependent roughness values estimated from the 10-min sonic observations. Orographic effects can thus be negligible at the mast (between sonics) but the effect of the sea-to-land roughness change upwind the fence is perhaps important. Inflow conditions derived from the sonics are related mostly to the flow characteristics upwind the closest sea-to-land roughness change; the wind profile is in equilibrium with the new surface



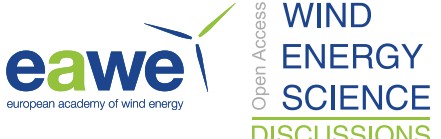

the first ≈1 m only. When evaluating flow models, topographic effects can be added. We provide the data to derive the inflow
conditions that we use to compute the speed-ups and so other inflow conditions can be used if preferred.

The speed-up follows the expected behavior; for increasing relative directions, the flow is less disturbed by the fence, and within the near-wake region, the porous fence has a lower effect on the flow than the solid fence. For model evaluation, the relative direction distribution needs to be taken into account, as its effects are well noticed. We observe a deeper effect of the fence on the flow in the stable compared to the near-neutral case with the same relative-direction interval; model comparison
is encouraged to distinguish if this is a result of stability or of the relative-direction distribution. For all cases, the fence slows the wind down for $z/h \leq 1.5$, and for some cases the fence speeds up the flow.

The velocity deficit profiles within the far-wake region ($x/h > 6.24$) converge onto each other. Counihan et al.'s solution agrees better with the self-preserving profiles than Perera's expression, which overestimates the effect of the fence on the flow at $z/h \leq 1$. This is mainly due to the low wake-momentum coefficient when compared to that used for 2D obstacles.
Counihan et al. and Perera's works are mostly based on wind-tunnel studies for flow nearly perpendicular to the obstacle. Model evaluation with our measurements could provide insights about accounting for 3D effects on analytical solutions and the wake-momentum coefficient dependency on relative directions.

*Acknowledgements.* Funding from EUDP, Denmark to both the IEA Task 27 'small turbines in high turbulence sites' and the 'Online WAsP' project (www.mywindturbine.com) are acknowledged.





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





**Figure 9.** Averaged speed-up $\langle \mathrm{sgn}(\widetilde{u})\, \widetilde{U}(z) \rangle_{\mathrm{ws}}/\langle U_o(z) \rangle$ on each vertical level behind the fence for a number of cases. $\pm$ the standard error is shown in the error bars





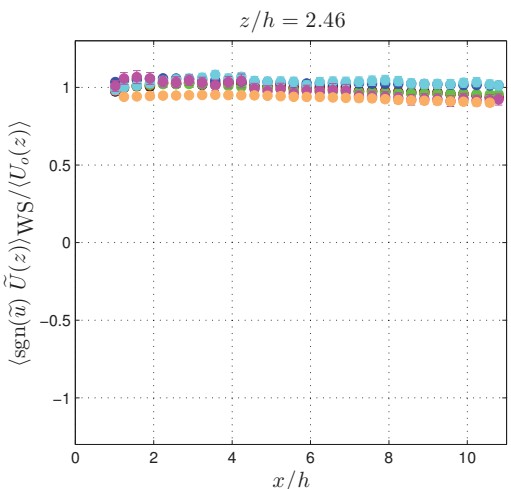

**Figure 10.** Same as Fig. 9 but for $z/h = 2.46$

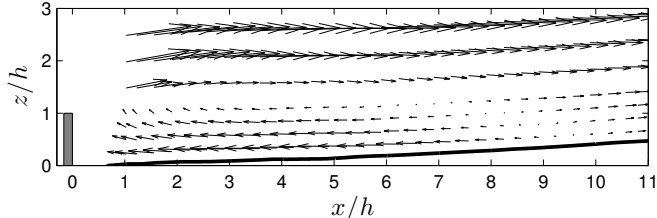

**Figure 11.** Velocity vector downwind the fence based on the CFD simulation for $\theta = 0°$





**Figure 12.** Self-preserving profiles for three cases and a number of downwind distances (details in the text). Results from Eqs. (8) and (9) are also shown