# Peer review of "The fence experiment – full-scale lidar-based shelter observations"

_Wind Energy Science, 2016_

## Referee Comment (RC1) · P. A. Taylor (Referee) · 2 May 2016

Comments on "The fence experiment – full-scale lidar-based shelter observations" by Alfredo Peña, Andreas Bechmann, Davide Conti, and Nikolas Angelou.

Peter A. Taylor, Centre for Research in Earth and Space Science, York University.

As the authors comment, there are relatively few field data on wakes behind fences or indeed behind other surface mounted obstacles, and the study reported by Peña et al (2016) is a valuable addition. They could however have noted the study by Wilson (2004) of flow behind a windbreak of porosity 0.45. One advantage of Wilson's study was a high aspect ratio (fence length (L)/fence height(h)) of 91.2 compared to only 10 in the Peña et al case. In order to consider the flow as approximately 2D for comparison with the Counihan et al (1974) theory it would be desirable to have a much longer fence

section. The measurements are relatively close to the fence (x/h <11 or x/L < 1.1) but pressure effects act in an isotropic manner and one might expect some departure from two dimensional behaviour.

The experimental situation is also complicated by the water to land roughness transition at the coastline about 78 m away from the fence. Checking Google Earth (my version of which still shows the fence in place) indicates virtually no step change of elevation at the waters edge. This is good, but there will be some impact from the roughness and possible heat flux changes within the internal boundary layer. The effective roughness lengths listed in Table 4, computed from friction velocity and wind speed measurements via Equations 1 and 2, are mostly of order 0.002 m. There is an error in Equn 2 (exponent 1/2 should be 1/4), which I assume is typographical, but the roughness lengths are lower than normally expected values for a grass covered surface ($\sim$0.01m). The flow at the measurement heights 6m and 12m will not have fully adjusted to the land surface change by the time it reaches the fence where the internal boundary layer depth in neutral conditions for flow normal to the shoreline would be $\sim$8m from simple guidelines, Walmsley et al (1989). Certainly the shear stresses at measurement heights will be low - more characteristic of the over water conditions - and lead to the computation of relatively low z0 values. Impacts on speed-up factors may however be small.

The triple Doppler lidar windscanner approach is an excellent remote sensing addition to in-situ measurements with cups or sonics but there are limitations in terms of sampling volume and turbulence measurements. The flow being measured in this instance would be fully accessible with masts and sonics and these could have provided an alternative or supplementary measurement technique. The direct application to wind turbine siting is rather limited since the measurements are restricted in height (z/h < 3) and downwind extent (x/h < 11). Additional turbulence measurements would have been useful for considerations of potential wear or breakdown for small wind turbines.

A minor suggestion related to Equations (10) and (11) would be that <ln(z0)> might be

a better option than ln<z0>.

The WEMOD and WASP-Shelter models are discussed with the comment that they are based on Counihan et al (1974) and 2-D models. While that is true in part there is also careful consideration of wakes behing surface mounted 3D obstacles in Taylor and Salmon's (1993) WEMOD model and 3D effects may play a role in the present study. A major uncertainty in applications of WEMOD is the estimated value used for the parameters Ch and Chtilde based on the drag and couple on the object. For an infinite 2D fence WEMOD suggests that Chtilde = 0.8 (1-$\varphi$) where $\varphi$ is the porosity (= 0 for the solid fence). However for a finite length fence section there is a suggestion that the coefficient (0.8) should be reduced (0.2 - 0.4) for "long low buidings". We have run both cases (0.8 and 0.4) for the10h length fence and for all flow directions.

Running WEMOD for the Peña et al solid fence and with z/h = 0.46 we get "speed-up" results, U(x,z)/U0(z), shown in Table I. The Peña et al results were extracted from Figure 9 of their paper, estimating values appropriate to cases I and II for flow normal to the fence, cases III and IV for flow at + 30° and -30° to normal (ignoring the slight asymmetry in the set-up) and case IV for flow at -60° to normal. WEMOD results are averages of calculations at 1° intervals within +/- 15° of the nominal direction. We set z0 = 0.002m as a representative value. WEMOD is a "far wake" model, intended primarily for x/h>6 but even that range can be optimistic for a solid 2D fence. Comparing estimates with Chtilde = 0.8 with the Peña et al measurements at z/h = 0.46 it is clear that the WEMOD model overestimates the wind speed reductions at all x locations while with Chtilde = 0.4 it generally underestimates them until x/h = 10 where, perhaps fortuitously, they match for all flow direction bins.

The data set for a porous fence ($\varphi$ = 0.375) is for flow at an angle of -30° to the normal to the fence (Case VII). At z/h = 0.46 we estimate the "speed-up" from Figure 9 in Peña et al to be 0.75 at x/h -= 10 and 0.45 at x/h = 6. The WEMOD model predicts less sheltering, even with Chtilde = 0.8, averaged over +/- 15° for this flow direction (from 30° to left of the upwind normal to the fence) and has corresponding speed-up values

of 0.89 and 0.63.

In contrast to these examples of poor results from WEMOD, Table II presents comparisons with the measurements reported by Wilson (2003). As with Table I there is averaging of WEMOD calculations over $1°$ values within $+/-15°$ of the wind directions indicated. There are sometimes a range of values extracted from Wilson's plots because of stability differences and the values in the table are intended to span neutral conditions. In general WEMOD values (with Chtilde = $0.8(1-\varphi)$ since the Wilson fence has L/h = 91.2 and we are looking at distances x/h < 20) are within or close to the range reported by Wilson. An exception is for x/h = 4 with flow normal and at $30°$ to the upwind normal to the fence but this is close to the fence and not in the far wake for which WEMOD application is anticipated.

There are differences between the plastic windbreak fencing used by Wilson and the porous wooden structure used in the Peña et al study, and in the different lengths of the two fences but porosities were similar. A comparison of speed-up values for three flow directions (normal, $+/- 30°$, $+/-60°$) at x/h = 10, z/h $\approx$ 0.5 shows 0.73, 0.78, 0.89 with porosity 0.45 from Wilson and 0.75, 0.9, 1.0 with porosity 0.375 from Peña et al. For flow normal to the fence these are compatible but for $30°$ and $60°$ angles the relatively short fence in the Peña et al study may allow flow around the ends which increases the speed-up.

In the near wake region there is relatively strong reverse flow (u < 0) but it is not clear to what extent the wind speed change is affected by the v component, especially for flow at $60°$ to the fence normal (Case IV, Fig 9, z/h = 0.21) where it appears that a vortex parallel to the fence may exist. Separate plots of u and v components, perhaps normalised by $U_0(h)$, in addition to Figure 9 could provide additional information.

p14 I am puzzled by the <$\sim$>ws combination since (p10) the $\sim$ symbol appears to indcate a sonic measurement over a time interval corresponding to a full WS scan. Also I assume U(z) should be U(x,z) in this context.

References

Counihan, J., Hunt, J. C. R., and Jackson, P. S., 1974, Wakes behind two-dimensional surface obstacles in turbulent boundary layers, J. Fluid Mech., 64, 529–563.

Peña, A., Bechmann, A., Conti, D. and Angelou, N., 2016, The fence experiment – full-scale lidar-based shelter observations, Wind Energ. Sci. Discuss., doi:10.5194/wes-2016-8

Taylor, P.A. and Salmon, J.R., 1993, A model for the correction of surface wind data for sheltering by upwind obstacles, J. Appl. Meteorol., 32, 1683–1694

Walmsley, J.L., Taylor, P.A. and Salmon, J.R., 1989: Simple Guidelines for Estimating Wind Speed Variations due to Small-Scale Topographic Features - An Update, Climatological Bulletin, 23(1), 3-14.

Wilson. J, 2004, Oblique, Stratified Winds about a Shelter Fence. Part I: Measurements, J. Appl. Meteorol., 43, 1149-1167.

Table I SHELCOR Comparisons with Peňa et al (2016) measurements; Short fence; $z/h$ = 0.46; z0 = 0.002m; h = 3m; Porosity 0.

Table II SHELCOR Comparisons with Wilson (2003) measurements; Long fence; $z/h$ = 0.50; z0 = 0.019m; h - 1.25m; Porosity 0.45.

Please also note the supplement to this comment:
http://www.wind-energ-sci-discuss.net/wes-2016-8/wes-2016-8-RC1-supplement.pdf

Table I      SHELCOR Comparisons with Peña et al (2016) measurements; Short fence; z/h = 0.46; $z_0$ = 0.002m; h = 3m; Porosity 0.

| x(m) | x/h | Peña et al Obs | | | SHELCOR 0.8 | | | SHELCOR 0.4 | | |
|---|---|---|---|---|---|---|---|---|---|---|
| | | 0° | 30° | 60° | 0° | 30° | 60° | 0° | 30° | 60° |
| 12 | 4 | -0.30 | 0.20 | 1.10 | -0.37 | -0.13 | 0.75 | 0.32 | 0.44 | 0.87 |
| 18 | 6 | 0.30 | 0.80 | 1.00 | 0.09 | 0.38 | 0.96 | 0.54 | 0.69 | 0.98 |
| 24 | 8 | 0.60 | 0.87 | 1.00 | 0.34 | 0.66 | 0.99 | 0.67 | 0.83 | 1.00 |
| 30 | 10 | 0.75 | 0.90 | 1.00 | 0.50 | 0.81 | 1.00 | 0.75 | 0.90 | 1.00 |

Table II      SHELCOR Comparisons with Wilson (2003) measurements; Long fence; z/h = 0.50; $z_0$ = 0.019m; h - 1.25m; Porosity 0.45.

| x(m) | x/h | Wilson Obs | | | SHELCOR 0.8(1-φ) | | |
|---|---|---|---|---|---|---|---|
| | | 0° | 30° | 60° | 0° | 30° | 60° |
| 5 | 4 | 0.44-0.52 | 0.50 | 0.63-0.67 | 0.19 | 0.30 | 0.63 |
| 7.5 | 6 | 0.46-0.58 | 0.60 | 0.75-0.79 | 0.49 | 0.57 | 0.78 |
| 12.5 | 10 | 0.64-0.74 | 0.73 | 0.84-0.87 | 0.73 | 0.78 | 0.89 |
| 18.75 | 15 | 0.80-0.86 | 0.83 | 0.91-0.93 | 0.84 | 0.87 | 0.91 |
| 26 | 20 | 0.87-0.94 | 0.88 | 0.90-0.92 | 0.89 | 0.91 | 0.96 |

**Fig. 1.** Tables

---

## Referee Comment (RC2) · E. S. Takle (Referee) · 10 May 2016

The strength of this paper is its reporting of measurements from a modern remote sensing system applied to a longstanding research problem of quantifying flow through and around porous obstacles, particularly fences and real (three dimensional) shelterbelts. Being a quasi-2-D study (analysis in vertical plane but quasi because of variation of wind direction perpendicular to the vertical plane), the contributions of this study to the literature on shelterbelt flow is limited, except the effects of thermal stratification. However, the evaluation of the WindScanner lidar system on the reasonably well-known flow field around fences is a useful contribution – the title appropriately captures this feature. And the care exercised by the authors in characterizing and understanding the inflow conditions in section 4.2 is very commendable and serves as a model for other such studies.

[Figure]

Some specific comments:

p. 7. "These three devices conform the WS": Do you mean "These three devices comprise the WS"?

p. 7. "The volume depends on the probe length of each lidar, which is considered to be twice the Rayleigh length zR. At the focused distances of 28 and 42 m, the lidars operate with zR = 0.67 and 1.52 m, respectively..." So are you implying that the volume is zR3 or (2zR)3 or something else? Please state the scanning volume in relation to the grid shown in Fig. 4.

Fig. 1. It should be clearly stated whether this is figure is a result of measurements or numerical simulation associated with this experiment or a conceptual view of what the flow field is envisioned to be behind a generic shelter of height h. And, of particular importance, if this image was adopted from another publication appropriate credit should be given.

The terminology of section 5.1 is confusing. The term "speed-up" suggests a definition of [U(z)-Uo(z)]/Uo(z), so that no change = a speed up of zero. But the caption of Fig. 7 defines speed-up as U(z)/Uo(z), which is more precisely defined as a wind speed ratio.

Fig. 8. caption: To be fully clear, the words "color bar" could be added to the first sentence, to read: "Average speed-up {....} (color bar) behind the fence..." And here, as above, it seems that wind speed ratio should be used in place of speed up.

Fig. 8 caption states that the "Vectors indicate the magnitude and direction of the ensemble-averaged u-component". This is more precisely stated as the "...magnitude and sign of the... u-component".

Figure 8 begs the question of what happens with v, the along-shelter component of the wind behind the shelter? No mention is made of v, which has important contributions to both mass conservation and practical "sheltering effects" such as protecting sensitive plants from damage or depositing snow. The along-shelter, v, component is quite

[Figure]

strong near the fence for a solid fence as has been shown in the shelterbelt literature, even for infinitely long fences. So it is inaccurate to state that the sheltering function is high if u is small but U is large. An example is p. 15 where the term "deepest shelter effect" is applied to the region of low u but where v and hence U might be large. Furthermore, v' contributes to turbulence that affects the u component as well.

The authors have missed an opportunity to make wider comparison of their work, particularly Figs. 9-12, with published results relating to shelterbelts. An example is the paper on measurements near fence of Wilson (2004) and papers cited therein on modeling of nornal and oblique flow to barriers. While the flow fields for neutral stratification in the vicinity of fences and thick shelters have been more widely published, few measurements are available of the effects of thermal stratification. The range of z/L in his study is rather small, but the results are important nevertheless.

In summary, the paper is a useful contribution in relation to addressing the measurement challenges of a modern wind field observing facility as revealed through measurements of a reasonably well-known flow field. Their results add modestly to the literature of flow fields in the vicinity of porous barriers, except the inclusion of thermal effects.

References Wilson, J. D., 2004: Oblique, stratified winds about a shelter fence. Part I: Measurements. J. Appl. Meteor., 43, 1149-1167.

———————————

---

## Author Comment (AC1) · 30 May 2016

We are very thankful for your comments because they are very helpful and constructive. Here follows our response item by item. The response is given as: xxx— response — xxx

As the authors comment, there are relatively few field data on wakes behind fences or indeed behind other surface mounted obstacles, and the study reported by Peña et al (2016) is a valuable addition. They could however have noted the study by Wilson (2004) of flow behind a windbreak of porosity 0.45. One advantage of Wilson's study was a high aspect ratio (fence length (L)/fence height(h)) of 91.2 compared to only 10 in the Peña et al case.

xxx— We were not aware of the study by Wilson (2004). The study is now referred to

in the revised version of the paper (first in the Introduction and later in the Conclusion and Discussion section)—xxx

In order to consider the flow as approximately 2D for comparison with the Counihan et al (1974) theory it would be desirable to have a much longer fence section. The measurements are relatively close to the fence (x/h <11 or x/L < 1.1) but pressure effects act in an isotropic manner and one might expect some departure from two dimensional behaviour.

xxx— We agree with the comment and we now add a sentence in the Conclusion and Discussion section related to this issue —xxx

The experimental situation is also complicated by the water to land roughness transition at the coastline about 78 m away from the fence. Checking Google Earth (my version of which still shows the fence in place) indicates virtually no step change of elevation at the waters edge. This is good, but there will be some impact from the roughness and possible heat flux changes within the internal boundary layer. The effective roughness lengths listed in Table 4, computed from friction velocity and wind speed measurements via Equations 1 and 2, are mostly of order 0.002 m.

xxx— We are aware of these issues and we mentioned them along the lines 12–14 in Page 11 and lines 29–30 in Page 13 of the original submission —xxx

There is an error in Equn 2 (exponent 1/2 should be 1/4), which I assume is typographical, but the roughness lengths are lower than normally expected values for a grass covered surface (~0.01m).

xxx— Yes, this is a typo and $\frac{1}{4}$ is indeed used in the calculations. The low roughness lengths are due to the influence of the upstream surface conditions —xxx

The flow at the measurement heights 6m and 12m will not have fully adjusted to the land surface change by the time it reaches the fence where the internal boundary layer depth in neutral conditions for flow normal to the shoreline would be ~8m from

simple guidelines, Walmsley et al (1989). Certainly the shear stresses at measurement heights will be low - more characteristic of the over water conditions - and lead to the computation of relatively low z0 values. Impacts on speed-up factors may however be small.

xxx— We agree with the reviewer's comment (see our statements in the original submission between lines 30—34 in Page 17 and lines 4—5 in Page 18) —xxx

The triple Doppler lidar windscanner approach is an excellent remote sensing addition to in- situ measurements with cups or sonics but there are limitations in terms of sampling volume and turbulence measurements. The flow being measured in this instance would be fully accessible with masts and sonics and these could have provided an alternative or supplementary measurement technique.

xxx— We agree with the reviewer and we believe that with the actual state of the WindScanner, it is still recommendable to supplement the measurements with those from cups and sonics. However, we didn't have the possibility to do so and we wanted to avoid distortion from the masts and booms both on the winds on the vertical plane and on the cups and sonics themselves (which seemed to be a concern in Wilson, 2004). It is also important to note that such flow is challenging for cup anemometers as their accuracy degrades with flow angle. We now also mention these challenges in the Conclusion and Discussion section —xxx

The direct application to wind turbine siting is rather limited since the measurements are restricted in height ($z/h < 3$) and downwind extent ($x/h < 11$).

xxx— We do not agree with the reviewer. A direct connection to turbine siting is made when, e.g., looking at small wind turbines. In some countries, like Denmark, small turbines (<25 kW) are normally installed close to buildings both vertically and horizontally ($z<25$ m and $x<20$ m). We now provide some more clear statements related to this in the Introduction of the revised version —xxx

Additional turbulence measurements would have been useful for considerations of potential wear or breakdown for small wind turbines.

xxx— This is definitively true and perhaps we should have also tried to perform turbulence measurements with the three lidars focusing at the same point for a long period to look at spectra characteristics. We now also mention this in the Conclusion and Discussion section of the revised version —xxx

A minor suggestion related to Equations (10) and (11) would be that <ln(z0)> might be a better option than ln<z0>.

xxx— We tested both options and there is no difference when estimating z_o. This is because <z_o> in Eqns. 10 and 11 is the result of averaging the z_o values in table 3 within the appropriate \theta interval for each case. For example for case 2 (0+-30 deg.) we have 6 z_o values (-25:5:25) deg. —xxx

The WEMOD and WASP-Shelter models are discussed with the comment that they are based on Counihan et al (1974) and 2-D models. While that is true in part there is also careful consideration of wakes behing surface mounted 3D obstacles in Taylor and Salmon's (1993) WEMOD model and 3D effects may play a role in the present study. A major uncertainty in applications of WEMOD is the estimated value used for the parameters Ch and Chtilde based on the drag and couple on the object. For an infinite 2D fence WEMOD suggests that Chtilde= 0.8 (1-$\varphi$) where $\varphi$ is the porosity (= 0 for the solid fence). However for a finite length fence section there is a suggestion that the coefficient (0.8) should be reduced (0.2 - 0.4) for "long low buidings". We have run both cases (0.8 and 0.4) for the 10h length fence and for all flow directions.

xxx— We agree with the reviewer and in the revised version we now add that both WEMOD and WAsP-shelter make considerations about 3D obstacles —xxx

Running WEMOD for the Peña et al solid fence and with z/h = 0.46 we get "speed-up" results, U(x,z)/U0(z), shown in Table I. The Peña et al results were extracted from

Figure 9 of their paper, estimating values appropriate to cases I and II for flow normal to the fence, cases III and IV for flow at + 30° and -30° to normal (ignoring the slight asymmetry in the set-up) and case IV for flow at -60° to normal. WEMOD results are averages of calculations at 1° intervals within +/- 15° of the nominal direction. We set z0 = 0.002m as a representative value. WEMOD is a "far wake" model, intended primarily for x/h>6 but even that range can be optimistic for a solid 2D fence. Comparing estimates with Chtilde = 0.8 with the Peña et al measurements at z/h = 0.46 it is clear that the WEMOD model overestimates the wind speed reductions at all x locations while with Chtilde = 0.4 it generally underestimates them until x/h = 10 where, perhaps fortuitously, they match for all flow direction bins.

The data set for a porous fence ($\varphi$ = 0.375) is for flow at an angle of -30° to the normal to the fence (Case VII). At z/h = 0.46 we estimate the "speed-up" from Figure 9 in Peña et al to be 0.75 at x/h -= 10 and 0.45 at x/h = 6. The WEMOD model predicts less sheltering, even with Chtilde = 0.8, averaged over +/- 15° for this flow direction (from 30° to left of the upwind normal to the fence) and has corresponding speed-up values of 0.89 and 0.63.

In contrast to these examples of poor results from WEMOD, Table II presents comparisons with the measurements reported by Wilson (2003). As with Table I there is averaging of WEMOD calculations over 1° values within +/-15° of the wind directions indicated. There are sometimes a range of values extracted from Wilson's plots because of stability differences and the values in the table are intended to span neutral conditions. In general WEMOD values (with Chtilde = 0.8(1-$\varphi$) since the Wilson fence has L/h = 91.2 and we are looking at distances x/h < 20) are within or close to the range reported by Wilson. An exception is for x/h = 4 with flow normal and at 30° to the upwind normal to the fence but this is close to the fence and not in the far wake for which WEMOD application is anticipated.

There are differences between the plastic windbreak fencing used by Wilson and the porous wooden structure used in the Peña et al study, and in the different lengths of the

two fences but porosities were similar. A comparison of speed-up values for three flow directions (normal, +/- 30°, +/-60°) at x/h = 10, z/h ≈ 0.5 shows 0.73, 0.78, 0.89 with porosity 0.45 from Wilson and 0.75, 0.9, 1.0 with porosity 0.375 from Peña et al. For flow normal to the fence these are compatible but for 30° and 60° angles the relatively short fence in the Peña et al study may allow flow around the ends which increases the speed-up.

xxx— We appreciate that Dr. Taylor has already made some comparisons of his model with the data. That is the main purpose of the paper. We would like to remind that the reader does not have to extract results from the figures because we provide the data at the weblinks, which are mentioned in the paper. Also and most important, we believe that when performing model evaluation, consideration of the direction distribution is needed as we have also seen that the results are very dependent on this. The direction distribution per case is also available at the weblinks. We are very much certain that if the evaluation is performed taking into account the direction distribution, the results of WEMOD will generally be closer to the observations —xxx

In the near wake region there is relatively strong reverse flow (u < 0) but it is not clear to what extent the wind speed change is affected by the v component, especially for flow at 60° to the fence normal (Case IV, Fig 9, z/h = 0.21) where it appears that a vortex parallel to the fence may exist. Separate plots of u and v components, perhaps normalised by U0(h), in addition to Figure 9 could provide additional information.

xxx— We thought about showing both components (u and v) but, in the original submission, we decided to concentrate on the horizontal wind speed only. However, we now have both u and v components as we also think they provide additional information —xxx

p14 I am puzzled by the <~>ws combination since (p10) the ~ symbol appears to indicate sonic measurement over a time interval corresponding to a full WS scan. Also I assume U(z) should be U(x,z) in this context

xxx— Page 10 states that $\sim$ corresponds to sonic statistics based on the time period that a full-scan is completed. Perhaps the problem is that we say "sonic statistics" and it should be just "statistics" no matter the instrument. This is now corrected in the revised version and U(x,z) is also now used when appropriate —xxx

―――――――――――――――――――――

---

## Author Comment (AC2) · 30 May 2016

We are very thankful for your comments because they are very useful and positive. Here follows our response item by item. The response is given as: xxx— response —xxx

The strength of this paper is its reporting of measurements from a modern remote sensing system applied to a longstanding research problem of quantifying flow through and around porous obstacles, particularly fences and real (three dimensional) shelter-belts. Being a quasi-2-D study (analysis in vertical plane but quasi because of variation of wind direction perpendicular to the vertical plane), the contributions of this study to the literature on shelterbelt flow is limited, except the effects of thermal stratification. However, the evaluation of the WindScanner lidar system on the reasonably well-known

flow field around fences is a useful contribution – the title appropriately captures this feature. And the care exercised by the authors in characterizing and understanding the inflow conditions in section 4.2 is very commendable and serves as a model for other such studies.

xxx— We thank the reviewer for the general feedback on our manuscript. The reviewer also states that "the contributions of this study to the literature are limited". We cannot find many papers where detailed fence-induced wake data are provided, for different fence types, and inflow conditions. Particularly, the data can be used directly to perform flow model evaluation, which is rather difficult to do with the actual literature due to lack of information about the instruments, the setups, the accuracy of the data (or the data themselves), and the description of the inflow conditions (a really important issue that the reviewer is also aware of). For example, in Wilson (2004) the concern regarding the accuracy of the measurements is evident. With respect to the later study our experiment provides e.g. measurements at four levels up to the fence height (1 level in Wilson, 2004) and at 27 downwind positions (4 in Wilson, 2004) within the range x/h<10, which is the region where models have more difficulties to predict the shelter —xxx

Some specific comments: p. 7. "These three devices conform the WS": Do you mean "These three devices comprise the WS"?

xxx— Yes, comprise is definitely a better word and so we use it now in the revised version —xxx

p. 7. "The volume depends on the probe length of each lidar, which is considered to be twice the Rayleigh length zR. At the focused distances of 28 and 42 m, the lidars operate with zR = 0.67 and 1.52 m, respectively..." So are you implying that the volume is zR3 or (2zR)3 or something else? Please state the scanning volume in relation to the grid shown in Fig. 4.

xxx— We rephrased the sentences to give a more accurate description of the measurement volume —xxx

Fig. 1. It should be clearly stated whether this is figure is a result of measurements or numerical simulation associated with this experiment or a conceptual view of what the flow field is envisioned to be behind a generic shelter of height h. And, of particular importance, if this image was adopted from another publication appropriate credit should be given.

xxx— We now explicitly say what this figure is —xxx

The terminology of section 5.1 is confusing. The term "speed-up" suggests a definition of $[U(z)-Uo(z)]/Uo(z)$, so that no change = a speed up of zero. But the caption of Fig. 7 defines speed-up as $U(z)/Uo(z)$, which is more precisely defined as a wind speed ratio.

xxx— We agree with the reviewer and so we now revised all instances where we mention speed up and now wind-speed ratio is used when appropriate —xxx

Fig. 8. caption: To be fully clear, the words "color bar" could be added to the first sentence, to read: "Average speed-up {....} (color bar) behind the fence..." And here, as above, it seems that wind speed ratio should be used in place of speed up.

xxx— The suggestion is taken into account in the revised version —xxx

Fig. 8 caption states that the "Vectors indicate the magnitude and direction of the ensemble-averaged u-component". This is more precisely stated as the "...magnitude and sign of the ... u-component".

xxx— The suggestion is taken into account in the revised version —xxx

Figure 8 begs the question of what happens with v, the along-shelter component of the wind behind the shelter? No mention is made of v, which has important contributions to both mass conservation and practical "sheltering effects" such as protecting sensitive plants from damage or depositing snow. The along-shelter, v, component is quite strong near the fence for a solid fence as has been shown in the shelterbelt literature,

even for infinitely long fences. So it is inaccurate to state that the sheltering function is high if u is small but U is large. An example is p. 15 where the term "deepest shelter effect" is applied to the region of low u but where v and hence U might be large. Furthermore, v' contributes to turbulence that affects the u component as well.

xxx— We thought about showing both components (u and v) but, in the original submission, we decided to concentrate on the horizontal wind speed only. However, we now have both u and v components as we also think they provide additional information —xxx

The authors have missed an opportunity to make wider comparison of their work, particularly Figs. 9-12, with published results relating to shelterbelts. An example is the paper on measurements near fence of Wilson (2004) and papers cited therein on modeling of normal and oblique flow to barriers. While the flow fields for neutral stratification in the vicinity of fences and thick shelters have been more widely published, few measurements are available of the effects of thermal stratification. The range of z/L in his study is rather small, but the results are important nevertheless.

xxx— We agree with the reviewer. Comparison with shelter measurements from the literature is now provided in the Conclusion and Discussion section —xxx

In summary, the paper is a useful contribution in relation to addressing the measurement challenges of a modern wind field observing facility as revealed through measurements of a reasonably well-known flow field. Their results add modestly to the literature of flow fields in the vicinity of porous barriers, except the inclusion of thermal effects.

---

## Author Response (AR1)

Reply to the Associate Editor comments:

Thank you very much for your comments. As we indicated in our reply to each of the reviewers we have revised the manuscript and implemented the changes when appropiate. Specifically for your knowledge:

- appropriately referring to other literature, such as the Wilson (2004) study and Wang et al. (2001) (reference below)

Yes, we do include the study of Wilson (2004) and refer to that of Wang et al. (2001).

- correcting typo in eqn. 2

Now corrected

- including both u and v components in Fig. 9

We have both components now

- Taylor's comments on page 14

Revised text due to the comment

- discussion of Fig. 1 (see Takle's comment)

Yes, now it is clear what it is

- discussion of the v component as suggested by Takle

As we now present the v-component we need to discuss these observations

Best regards,

The authors

Summary
17-06-2016 10:13:56

Differences exist between documents.

| **New Document:** | **Old Document:** |
|---|---|
| template_rev | template |
| 25 pages (1000 KB) | 22 pages (929 KB) |
| 17-06-2016 10:13:11 | 17-06-2016 10:13:10 |
| Used to display results. | |

Get started: first change is on page 1.

No pages were deleted

**How to read this report**

Highlight indicates a change.
Deleted indicates deleted content.
▲ indicates pages were changed.
↔ indicates pages were moved.

[revised manuscript text omitted]

---

## Author Response (AR2)

Dear WES journal,

This is our response to the minor comments of the associate editor (response in italics):

1. Abstract line 2: Replace "sonic" with "sonic anemometer" (starting in the abstract, but also throughout)

*Changed throughout the manuscript as suggested*

2. abstract line 5 "the lower is the shelter" should be replaced with "the lower the effect of the shelter"

*Changed as suggested*

3. abstract line 6: "the lower the sheltering effect"

*Replaced by "no sheltering effect is observed" instead*

4. abstract line 8: insert "downwind" so sentence reads "up to 11 fence heights downwind"

*Changed as suggested*

5. page 2, line 1: no comma needed after "simulations"

*Changed as suggested*

6. page 2, line 3: insert "of" between "downwind" and "porous"

*Changed as suggested*

7. Literature review could cite other work, particularly one or more of the Wang and Takle works listed below, as the role of obliquity is nicely addressed in their simulations.

*Now there are two papers listed from Wang and Takle's studies*

8. page 4, line 14, please provide a reference for the pressure measurements on blocks in shear flows.

*Added as suggested (in fact it is described in Counihan et al.)*

9. throughout: the authors sometimes use past tense ("was measured" and "was used to simulate),

frequently present tense ("system is not", "mast is deployed", "sonic anemometers are placed" "we redefine"), and sometimes an implied future tense ("where we want to measure"). Please consider carefully ensuring consistency throughout the manuscript to enhance readability

*We are now using past tense for the events that really occurred in the past, like the measurements and the campaign, whereas the analysis is written in present tense. We now avoid the implied future tense.*

10. page 7 line 7: each of the windscanner lidars can only measure LOS speed. Please clarify in this sentence that the integration of the three measurements enable measurement of the wind direction (as discussed in 3.3.2), not that each individual lidar can define wind speed and wind direction.

*We change "direction" to "sign" because this is what we meant*

11. page 8 line 12: do you mean interference of the fence with the probe volume? I doubt the probe volume affects the fence!

*Yes, you are right. The sentence is now "reversed"*

12. section 3.3.1 should include relevant WS facts such as maximum and minimum detectable velocities

*We now include some relevant facts as suggested*

13. page 10 lines 3 and 4: should explicitly state that this procedure assumes w is zero

*This is now explicitly stated*

14. page 14 footnote: "take" should be "takes"

*Changed as suggested*

15. page 15 line 6: "is not uniformly distributed" rather than "does not uniformly distribute"

*Changed as suggested*

16. page 15 line 22: "that causes", not "what causes"

*Changed as suggested*

17. page 16: first partial paragraph: please proofread and rewrite where necessary to ensure that all ")" have mates. Are so many parentheticals necessary?

*The whole paragraph is now revised and most of the () are avoided*

18. page 17 line 20: include reference to equation number for Perera's estimates as it has been several pages since Perera has been referred to.

*We now add "Perera's expression in Eq. (8)" one-two lines above the sentence to remind the reader of the expression and its origin*

19. page 17 bottom/ page 18 top: Should refer to website where data is available for interested readers. perhaps page 18 line 12?

*We now include the website address in the conclusions when appropriate*

20. page 18 line 16: perhaps "obvious" rather than "well noticed"?

*Changed as suggested*